# DualRes: A Resampling-Based Framework for Enhancing Probabilistic Forecasting

## Abstract

Probabilistic forecasting of time series has gained increasing attention in practice due to the need for assessing risks and uncertainties in future observations. In this manuscript, we propose DualRes, a framework that improves the probabilistic forecasting performance of existing algorithms by incorporating conditional heteroskedasticity and residual distributional information. Specifically, during training, DualRes employs two separate models to learn the conditional mean and volatility of the time series, while during inference it generates pseudo-normalized residuals through resampling. DualRes requires only mean forecasts, so it offers substantial flexibility in the choice of forecasting algorithms—even algorithms originally designed for mean forecasting can be adapted to probabilistic forecasting. DualRes applies to both univariate and multivariate time series and remains robust under non-Gaussian errors with conditional heteroskedasticity. Numerical experiments on six real-world datasets demonstrate its good empirical performance in capturing distribution of future observations and producing accurate prediction intervals.

## 1 Introduction

Time series is a common data type in real-world applications such as finance, energy management, and weather forecasting. After collecting a sequence of time series data, this manuscript focuses on probabilistic forecasting, which aims to predict the probability distribution of future observations and thereby support risk assessing and decision-making, as discussed in Luo et al. (2018); Nguyen & Quanz (2021); Wu & Politis (2024); Zheng et al. (2025) and the references therein.

To our knowledge, two types of methods are commonly considered in probabilistic forecasting. The first type, such as the work of Kollovieh et al. (2023); Chen et al. (2024b;a); Tashiro et al. (2021); Zheng et al. (2025), leveraged diffusion process and generative model, like those of Song et al. (2020); Ho et al. (2020); Kollovieh et al. (2025), to perform probabilistic forecasting. The validity of such methods in general relied on the assumption of time series having Gaussian distribution. Another stream that addressed probabilistic forecasting problems involved adjusting the training processes. Notable examples include Le Guen & Thome (2020); Rasul et al. (2021b); Hasson et al. (2021); Bergsma et al. (2023); Ansari et al. (2024). A common issue of these methods is that the underlying mathematical models and mechanisms of their validity are not transparent and rigorous to practitioners compared to those of diffusion model-based approaches.

In this manuscript, motivated by recent advances in bootstrap and resampling methods for statistical inference and prediction in time series analysis Wu & Politis (2024; 2025); Zhang et al. (2024), we propose DualRes, a resampling-based framework for probabilistic forecasting of time series data. DualRes consists of three steps. First, we train a predictive model—such as those in Zeng et al. (2023); Lin et al. (2024)—to estimate the conditional mean of the time series, and compute fitted residuals as the difference between the observations and the predictive means. Second, we introduce another model to estimate the conditional volatility, and normalize the fitted residuals by dividing them by the predicted volatilities. Finally, we apply bootstrap algorithms (see Efron (1979)) to resample the normalized residuals, and combine the estimated conditional mean and volatility to generate predictive distributions of future observations. As demonstrated in Wu (1986); Stine (1985); Chwialkowski et al. (2014), a well-designed bootstrap algorithm can approximate the underlying probability distribution of future time series without imposing restrictive distributional

assumptions, such as Gaussianity. Thus, DualRes relaxes the reliance on Gaussian distributions of diffusion process-based methods.

In addition to relaxing the Gaussian assumption, DualRes offers several advantages. First, it is flexible in the choice of conditional mean and volatility models. As shown Section 4.1, by applying a logarithmic transformation to the squared residuals, DualRes requires only mean forecasts to perform probabilistic forecasting. This allows models originally designed for mean forecasting to be adapted for probabilistic forecasting. Second, DualRes explicitly accounts for conditional heteroskedasticity and non-Gaussianity, thereby improving the performance of probabilistic forecasting methods that ignore these features. Finally, as established in Theorem 1, DualRes incorporates spatial dependence by resampling residual vectors, making it adaptable to multivariate time series settings.

We summarize the advantages of the proposed method as follows.

- **No Gaussianity assumption:** Our work does not rely on maximizing likelihood functions, so the data distributions are not necessarily Gaussian.
- **Flexibility in selecting mean/volatility forecasting algorithms:** Implementation of our work only needs models generating mean forecasts, thus offering good flexibilities.
- **Theoretical justification:** The validity of our approach stems from its ability to simulate the underlying data-generating process of time series instead of a black-box model. Furthermore, under some conditions, the resampling mechanism is ensured to capture the underlying distribution of innovations.
- **Robustness to conditional heteroskedasticity and multivariate Settings:** DualRes is adaptable for conditional heteroskedastic time series, and it accounts for spatial dependence in predictions.

## 2 RELATED WORKS

This work is related to the area of probabilistic time series forecasting and resampling. We provide a brief introduction of the latest studies for each area. In addition, we introduce the setting of probabilistic forecasting to make the manuscript self-contained.

**Probabilistic time series forecasting.** Diffusion models and their variants, like those introduced in Ho et al. (2020), have been applied to both univariate and multivariate probabilistic forecasting of time series Rasul et al. (2021a;b); Li et al. (2022); Chen et al. (2024b;a); Kollovieh et al. (2025); Zheng et al. (2025). By modeling time series data as a Markov chain with Gaussian transitions, these methods offer good interpretability in the training and inference stage. The state space model is another frequently used model that offers good interpretability and empirical performance. Recent works such as Rangapuram et al. (2018); Li et al. (2019) leveraged deep learning to describe parameters in the state space model. We also refer Rangapuram et al. (2021); Feng et al. (2024); Ansari et al. (2024) for other deep learning-based approaches to probabilistic forecasting.

**Resampling and bootstrap.** Bootstrap algorithm is a well-recognized method to quantify uncertainty of statistics, and has been employed to various fields of machine learning, like those in White & White (2010); Austern & Syrgkanis (2021); Shin et al. (2021); Rohekar et al. (2018); Wang et al. (2024b); Yu et al. (2024).

## 3 RESAMPLING ASSISTED PROBABILISTIC FORECASTING (DUALRES)

Suppose we observe a time series $\mathbf{x}_{1:T} \in \mathbf{R}^d$, with $t = 1, \cdots, T$ denoting the time steps. Our objective is to forecast the distributions of future observations $\mathbf{x}_{T+j}$ for $j = 1, 2, \cdots, J$. There have been discussions in the literature like Salinas et al. (2020) and Kollovieh et al. (2025). When further investigating these works, we find that they effectively incorporated the conditional mean and conditional volatility information in forecasting. However, these works commonly assigned a Gaussian distribution to the residuals, making the validity of forecasting algorithms rely on the residuals (and therefore, observations) obeying Gaussian distributions.

Our objective is to take into account the distributional information and avoid the assumption of Gaussian distribution in forecasting. To achieve the goal, we incorporate a resampling step into the

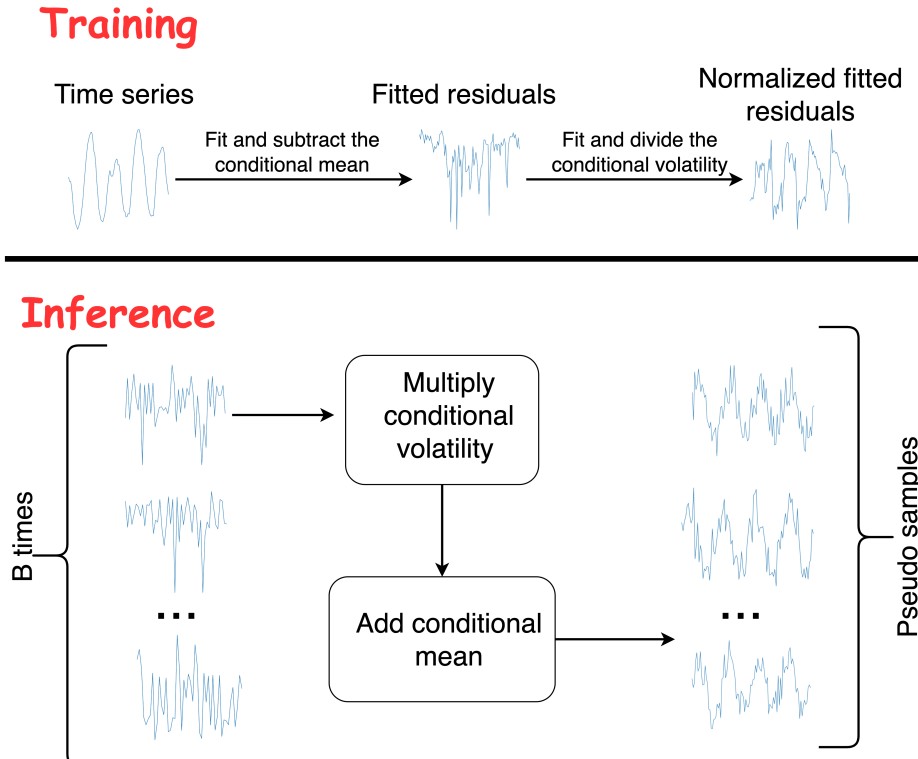

Figure 1: Structure of the training and inference stage.

forecasting algorithm 2. Resampling has been well employed in the literature such as Pan & Politis (2016), Wu & Politis (2025), and Zhang et al. (2025) in forecasting. However, to our knowledge, they did not account for the conditional heteroskedasticity (i.e., dependence of future variance on past observations), while our work allows for the existence of conditional heteroskedasticity in future observations.

### 3.1 TRAINING STAGE

Figure 1 presents an overview about the structure of the training and inference of stage of the proposed method. Our work is motivated by a two-stage conditional heterogeneous vector autoregressive model

$$\mathbf{x}_t = F(\mathbf{x}_{t-1}, \cdots, \mathbf{x}_{t-q}) + \boldsymbol{\zeta}_t, \quad \text{and} \quad \boldsymbol{\zeta}_t = G(\boldsymbol{\zeta}_{t-1}, \cdots, \boldsymbol{\zeta}_{t-s})\boldsymbol{\eta}_t, \tag{1}$$

where

$$G(\boldsymbol{\zeta}_{t-1}, \cdots, \boldsymbol{\zeta}_{t-s}) = \text{diag}\left(G_1(\boldsymbol{\zeta}_{t-1}, \cdots, \boldsymbol{\zeta}_{t-s}), \cdots, G_d(\boldsymbol{\zeta}_{t-1}, \cdots, \boldsymbol{\zeta}_{t-s})\right)$$

is a $d \times d$ diagonal matrix, $F : \mathbf{R}^{d \times q} \to \mathbf{R}^d$, $G_i : \mathbf{R}^{d \times s} \to [0, \infty)$ are functions to learn, and $\boldsymbol{\eta}_t$ are independent of past observations $\mathbf{x}_{-t}$ and $\boldsymbol{\zeta}_{-t}$, $\mathbf{E}\left[\boldsymbol{\eta}^{(t)}\right] = 0$, and $\boldsymbol{\eta}^{(t)}$ have identical distribution.

The functions $F$ and $G$ respectively controls the conditional mean and conditional volatility of time series data, Furthermore, such model offers a good property that the residual terms $\boldsymbol{\zeta}_t$ does not incur bias to the conditional mean $F$, which motivates the two-stage training procedure as in Algorithm 1. We prove this property in Section 4.

---

**Algorithm 1** Training a heterogeneous vector autoregressive model

---

**Require:** Time series data $\{\mathbf{x}_t : t = 1, \cdots, T\}$, lag $q$ for the conditional mean model, and lag $s$ for the conditional volatility model.

1: Train the conditional mean model $\widehat{F}$ and derive the fitted residuals

$$\widehat{\boldsymbol{\zeta}}_t = \mathbf{x}_t - \widehat{F}(\mathbf{x}_{t-q}, \cdots, \mathbf{x}_{t-1})$$

for $t = q + 1, \cdots, T$.

2: Train the conditional volatility model $\widehat{G}$ with the fitted residuals $\widehat{\boldsymbol{\zeta}}_t, t = q + 1, \cdots, T$. After that, derive the normalize fitted residuals

$$\widehat{\boldsymbol{\eta}}_t = \widehat{G}^{-1}\left(\widehat{\boldsymbol{\zeta}}_{t-s}, \cdots \widehat{\boldsymbol{\zeta}}_{t-1}\right)\widehat{\boldsymbol{\zeta}}_t, \tag{2}$$

where $t = q + s + 1, \cdots, T$.

---

**Remark 1.** *Practitioners may resort to mean forecasting methods, such as Lin et al. (2024), to establish the model $\widehat{F}$ for the conditional mean function $F$ in equation 1. Learning $G$, on the other hand, is not straightforward. After calculating $\widehat{\boldsymbol{\zeta}}_t$, this manuscript performs the transformation $\widehat{\boldsymbol{\iota}}_t = R(\widehat{\boldsymbol{\zeta}}_t)$ for $t = q + 1, \cdots, T$, where $R : \mathbf{R}^d \to \mathbf{R}^d$ is a function of the form:*

$$R(\mathbf{x}) = (\log(\mathbf{x}_1^2), \log(\mathbf{x}_2^2), \cdots, \log(\mathbf{x}_d^2))^\top \quad \text{and} \quad \mathbf{x} \in \mathbf{R}^d. \tag{3}$$

*We then use mean forecasting methods (e.g., those in Lin et al. (2024)) to learn $U_i = \log(G_i)$. We demonstrate in Section 4.1 that, despite taking logarithm transformations incur a constant bias when learning $\log(G_i)$, the constant bias will be self-eliminated during the normalization step equation 2 of Algorithm 1 and the sampling step equation 4 of the inference Algorithm 2. Consequently, the bias introduced during the training stage does not affect the prediction.*

The motivation of the model equation 1 originates from the ARMA-GARCH model, like those in Ling & McAleer (2003), that adopted linear models for both $F$ and $G$. The conditional heteroskedasticity considered in this manuscript associates the volatility with past observations, and is different from Ye et al. (2025), where the volatility was associated with exogenous features.

The flexibility of Algorithm 1 is reflected by its selection of models used to learn $F$ and $G$—mean forecasting algorithms, such as those proposed in Zeng et al. (2023); Zhang & Yan (2023); Lin et al. (2024), among others—can be employed to fulfill this purpose.

## 3.2 INFERENCE STAGE

The intuition behind Algorithm 2 involves simulating the data generating process in equation 1. If $\widehat{F}$ and $\widehat{G}$ closely approximate the true conditional mean $F$ and conditional volatilities $G$, then Theorem 1 in Section 4 guarantees that the distribution of the simulated normalized residuals $\boldsymbol{\eta}_j^*$ closely matches the distribution of the true normalized residuals $\boldsymbol{\eta}_j$. Furthermore, the generation of $\mathbf{x}_{T+j}^*$ follows the same autoregressive iteration as in equation 1. Therefore, under the assumption that equation 1 accurately characterizes the data generating process of $\mathbf{x}_t$, since the estimated conditional mean $\widehat{F}$, conditional volatility $\widehat{G}$, the distribution of pseudo-normalized residuals $\boldsymbol{\eta}_j^*$, and the autoregressive iteration all provide good approximations to that of $\mathbf{x}_t$, the distribution of the pseudo-samples $\mathbf{x}_{T+j}^*$ should be close to that of the actual future observations $\mathbf{x}_{T+j}$.

---

**Algorithm 2** Inference Stage

---

**Require:** Time series data $\mathbf{x}_{1:T}$, lag $q$ for conditional mean, lag $s$ for conditional volatility, prediction step $J$, resampling time $B$.

1: Derive the functions $\widehat{F}$ and $\widehat{G}$, as well as the normalized fitted residuals $\widehat{\boldsymbol{\eta}}_t$ as in Algorithm 1.
2: **for** $b \leftarrow 1$ to $B$ **do**
3:     Sample $\boldsymbol{\eta}_j^*$ for $j = 1, \cdots, J$ by drawing from $\widehat{\boldsymbol{\eta}}_{q+s+1}, \cdots, \widehat{\boldsymbol{\eta}}_T$ with replacement.
4:     Generate pseudo-samples $\mathbf{x}_{T+1}^*, \cdots, \mathbf{x}_{T+j}^*$ using the following iteration:

$$
\begin{aligned}
\boldsymbol{\zeta}_{T+j}^* &= \widehat{G}(\widehat{\boldsymbol{\zeta}}_{T+j-s}^*, \cdots, \widehat{\boldsymbol{\zeta}}_{T+j-1}^*)\boldsymbol{\eta}_j^*, \\
\mathbf{x}_{T+j}^* &= \widehat{F}(\mathbf{x}_{T+j-q}^*, \cdots, \mathbf{x}_{T+j-1}^*) + \boldsymbol{\zeta}_{T+j}^*,
\end{aligned}
\tag{4}
$$

     where $\mathbf{x}_{T+j-q}^* = \mathbf{x}_{T+j-q}$ and $\widehat{\boldsymbol{\gamma}}_{T+j-s}^* = \widehat{\boldsymbol{\gamma}}_{T+j-s}$ if $q, s \geq j$.
5: **end for**
6: For any measurable set $A \subset \mathbf{R}^{d \times J}$, we estimate the joint distribution of $\mathbf{x}_{(T+1):(T+J)}$ by the empirical measure $\frac{1}{B} \sum_{b=1}^B \mathbf{1}_{\mathbf{x}_{(T+1):(T+J)}^* \in A}$

---

**Remark 2.** *Practitioners may resort to Remark 1 to learn $G$. In such case, the value of $\widehat{G}(\widehat{\boldsymbol{\zeta}}_{T+j-s}^*, \cdots, \widehat{\boldsymbol{\zeta}}_{T+j-1}^*)$ can be derived through applying the learned autoregressive model to $\widehat{\boldsymbol{\iota}}_{T+j-s}^*, \cdots, \widehat{\boldsymbol{\iota}}_{T+j-1}^*$, where $\widehat{\boldsymbol{\iota}}_k^* = R(\boldsymbol{\zeta}_k^*)$.*

## 4 THEORETICAL JUSTIFICATION

The theoretical justification of DualRes is divided into two parts. First, we provide illustrations on why Algorithm 1 is capable of learning $F$ and $G$. After that, we summarize in Theorem 1 that the distribution of the pseudo-normalized residuals $\boldsymbol{\eta}_j^*$ closely approximates that of the true normalized residuals $\boldsymbol{\eta}_j$.

### 4.1 FURTHER DISCUSSIONS ON SECTION 3

To illustrate why the two-stage procedure in Algorithm 1 learns $F$ and $G$, from the tower property of conditional expectation,

$$
\begin{aligned}
\mathbb{E}\left[\boldsymbol{\zeta}_t \mid \mathbf{x}_{(t-q):(t-1)}\right] &= \mathbb{E}\left[\mathbb{E}\left[G(\boldsymbol{\zeta}_{t-1}, \cdots, \boldsymbol{\zeta}_{t-s})\boldsymbol{\eta}_t \mid \mathbf{x}_{(t-q):(t-1)}, \boldsymbol{\zeta}_{(t-s):(t-1)}\right] \mid \mathbf{x}_{(t-q):(t-1)}\right] \\
&= \mathbb{E}\left[(G(\boldsymbol{\zeta}_{t-1}, \cdots, \boldsymbol{\zeta}_{t-s})\mathbb{E}\boldsymbol{\eta}_t) \mid \mathbf{x}_{(t-q):(t-1)}\right] = 0.
\end{aligned}
$$

Therefore, when we train $\widehat{F}$, the residuals $\boldsymbol{\zeta}_t$ do not incur bias to $F$, making it possible for the estimator $\widehat{F}$ to closely approximate $F$. On the other hand, define the function $R$ as in equation 3, define $\boldsymbol{\gamma}_t = R(\boldsymbol{\zeta}_t)$, then the $i$-th element of $\boldsymbol{\gamma}_t$ is

$$
\gamma_{t,i} = \log\left(G_i^2(\boldsymbol{\zeta}_{t-1}, \cdots, \boldsymbol{\zeta}_{t-s})\right) + \log\left(\eta_{t,i}^2\right).
\tag{5}
$$

Furthermore, by assuming that the functions $G_i^2(\cdot), i = 1, \cdots, d$, depend on $\boldsymbol{\zeta}_{t-1}, \cdots, \boldsymbol{\zeta}_{t-s}$ only through their element-wise squares, and notice that $\zeta_{t,i}^2 = \exp(\gamma_{t,i})$, equation 5 implies that

$$
\boldsymbol{\gamma}_t = A(\boldsymbol{\gamma}_{t-1}, \cdots, \boldsymbol{\gamma}_{t-s}) + \boldsymbol{\iota}_t,
\tag{6}
$$

where $A : \mathbf{R}^{d \times s} \to \mathbf{R}^d$ is a function such that $A_i(\boldsymbol{\gamma}_{t-1}, \cdots, \boldsymbol{\gamma}_{t-s}) = \log\left(G_i^2(\boldsymbol{\zeta}_{t-1}, \cdots, \boldsymbol{\zeta}_{t-s})\right) + \mathbb{E}\left[\log\left(\eta_{t,i}^2\right)\right]$ and $\iota_{t,i} = \log\left(\eta_{t,i}^2\right) - \mathbb{E}\left[\log\left(\eta_{t,i}^2\right)\right]$. Therefore, the representation equation 6 allows the use of a mean-forecasting algorithm to learn $B$, which inevitably incurs a constant bias term $\mathbb{E}\left[\log\left(\eta_{t,i}^2\right)\right]$.

Fortunately, the constant bias does not affect the prediction as it self-eliminated during equation 2 of Algorithm 1, which divides the fitted residuals $\widehat{\boldsymbol{\zeta}}_t$ by $\widehat{G}$, and equation 4 of Algorithm 2, which multiplies the sampled $\boldsymbol{\eta}_j^*$ by $\widehat{G}$.

We would like to stress that the assumption of $G_i^2$ depending on $\boldsymbol{\zeta}_{t-1}, \cdots, \boldsymbol{\zeta}_{t-s}$ through their element-wise squares is common in the literature. For example, the ARMA-GARCH models in Ling

& McAleer (2003) leveraged this assumption. The advantage of this transformation is, by replacing $\gamma_t$ with $\widehat{\gamma}_t = R(\widehat{\zeta}_t)$, $\widehat{\gamma}_t$ approximately follows an additive autoregressive process equation 6, allowing the use of various conditional mean forecasting methods—such as those in Lin et al. (2024)—for estimating the function $A$ in equation 6.

## 4.2 VALIDITY OF THE RESAMPLE PROCEDURE

While conditional mean and volatility information has been widely leveraged in various probabilistic forecasting algorithms, like Salinas et al. (2020); Zheng et al. (2025), the distributional information of residuals $\eta_t$ has received comparatively less attention. Compared to directly assigning normal distribution to $\eta_t$, we introduce the resampling step equation 4 in Algorithm 2 to learn underlying distribution of $\eta_t$.

Furthermore, as illustrated in Section 3, the validity of Algorithm 2 comes from simulating the underlying data generating process of $\mathbf{x}_t$. Therefore, if model eq.equation 1 holds true and Algorithm 1 generates good estimators for $F$ and $G$ (up to a constant scale), the validity of Algorithm 2 is achieved provided that the empirical process of the vector $\widehat{\eta}_t$—characterized by the probability measure defined by the following joint cumulative distribution function (CDF in abbreviation)

$$\widehat{P}(\mathbf{y}) = \frac{1}{T - q - s} \sum_{t=s+q+1}^{T} \mathbf{1}_{\widehat{\eta}_t \leq \mathbf{y}} \tag{7}$$

where $\mathbf{1}_{\widehat{\eta}_t \leq \mathbf{y}}$ denotes for $\prod_{i=1}^{d} \mathbf{1}_{\widehat{\eta}_{t,i} \leq \mathbf{y}_i}$, converges to the distributions of $\eta^{(t)}$. Theorem 1 provides a theoretical justification for this claim.

**Theorem 1.** *Suppose $\eta_t, t = 1, 2, \cdots$, are independent and identical distributed. In addition, suppose conditions detailed in Section A of Appendix hold true. Then we have*

$$\sup_{\boldsymbol{y} \in \mathbf{R}^d} |\widehat{P}(\boldsymbol{y}) - P(\boldsymbol{y})| \to_p 0, \tag{8}$$

*where $\to_p$ denotes convergence in probability, $P(\cdot)$ denotes the CDF of $\eta_t$, and the convergence is with respect to the sample size $T \to \infty$.*

*Proof.* Postponed to Section A in Appendix. □

Theorem 1 guarantees that the distribution of the resampled normalized residuals $\eta_{t,i}^*$ in Algorithm 2 matches that of the true normalized residuals $\eta_{t,i}^*$. As a result, Algorithm 2 effectively captures the distributional information of $\eta_{t,i}^*$.

**Remark 3.** *According to Politis et al. (1999), sampling with replacement from $\widehat{\eta}_t$ is equivalent to drawing from the distribution with CDF $\widehat{P}(\cdot)$ as defined in e.q. equation 7. Therefore, the distribution of $\eta_i^*$ is guaranteed to match the distribution of $\eta_i$ once e.q. equation 8 is satisfied.*

## 5 NUMERICAL EXPERIMENTS

This section demonstrates the effectiveness of DualRes as a boosting algorithm for enhancing the performance of existing methods in both univariate and multivariate probabilistic forecasting. Due to the space limitations, the detailed experimental setup and additional experimental results—including hyperparameter choices, introduction of datasets and evaluation metrics, and demonstration of mean forecasting performance—are deferred to Section B of the Appendix.

### 5.1 UNIVARIATE PROBABILISTIC FORECASTING

**Dataset and experimental settings.** We run the experiments on six real-world commonly used time series dataset, respectively named *ETTh1, ETTh2, Electricity, Traffic, Exchange, and M4-Hourly.* The details about these datasets are introduced in Section B.1 of the Appendix.

The evaluation metrics are CRPS and MAEC (mean absolute error of coverage). A detailed introduction to these metrics is provided in Section B.2 of the Appendix. In addition to probabilistic

Table 1: Numerical experiment results on univariate time series datasets. The numbers in brackets indicate 95% confidence intervals, computed from five independent repetitions of each experiment. In the ablation studies, the better result is highlighted in bold, corresponding to smaller metric values, or, when metrics are equal, to narrower confidence intervals.

| Models | Metrics | ETTh1 | ETTh2 | Electricity | Traffic | Exchange | M4-Hourly |
|---|---|---|---|---|---|---|---|
| DeepAR | CRPS | 0.178(0.031) | **0.076(0.015)** | 0.082(0.001) | **0.107(0.007)** | 0.015(0.001) | 0.087(0.092) |
|  | MAEC | 0.411(0.082) | 0.394(0.148) | 0.454(0.001) | **0.443(0.035)** | 0.498(0.003) | 0.411(0.099) |
| DeepAR +Ours | CRPS | **0.176(0.011)** | 0.085(0.002) | **0.071(0.001)** | 0.115(0.003) | **0.010(0.001)** | **0.042(0.003)** |
|  | MAEC | **0.408(0.018)** | **0.393(0.021)** | **0.439(0.006)** | 0.471(0.013) | **0.466(0.026)** | **0.378(0.003)** |
| DLinear | CRPS | **0.185(0.001)** | 0.075(0.003) | 0.061(0.007) | **0.131(0.002)** | 0.019(0.008) | 0.048(0.005) |
|  | MAEC | 0.414(0.014) | 0.462(0.018) | 0.382(0.016) | 0.433(0.012) | **0.447(0.024)** | **0.373(0.020)** |
| DLinear +Ours | CRPS | 0.196(0.008) | **0.070(0.004)** | **0.054(0.001)** | 0.133(0.002) | **0.010(0.001)** | **0.040(0.012)** |
|  | MAEC | **0.388(0.013)** | **0.395(0.069)** | **0.367(0.007)** | **0.393(0.003)** | 0.465(0.011) | 0.409(0.016) |
| PatchTST | CRPS | **0.169(0.005)** | **0.066(0.010)** | 0.063(0.003) | **0.124(0.001)** | 0.013(0.003) | **0.041(0.006)** |
|  | MAEC | 0.431(0.013) | 0.406(0.076) | 0.375(0.013) | 0.435(0.013) | 0.475(0.037) | **0.386(0.056)** |
| PatchTST +Ours | CRPS | 0.200(0.043) | 0.073(0.001) | **0.063(0.001)** | 0.134(0.003) | **0.012(0.002)** | 0.056(0.024) |
|  | MAEC | **0.403(0.028)** | **0.399(0.089)** | **0.372(0.015)** | **0.413(0.002)** | **0.473(0.022)** | 0.416(0.027) |
| TimeMixer | CRPS | 0.365(0.005) | 0.095(0.004) | 0.273(0.006) | 0.384(0.001) | 0.027(0.008) | **0.107(0.012)** |
|  | MAEC | 0.415(0.006) | **0.383(0.004)** | 0.427(0.001) | 0.411(0.024) | 0.500(0.000) | 0.441(0.041) |
| TimeMixer +Ours | CRPS | **0.348(0.018)** | **0.094(0.001)** | **0.237(0.002)** | **0.356(0.001)** | **0.014(0.001)** | 0.144(0.018) |
|  | MAEC | **0.396(0.014)** | 0.429(0.006) | **0.400(0.001)** | **0.410(0.003)** | **0.421(0.076)** | **0.370(0.008)** |

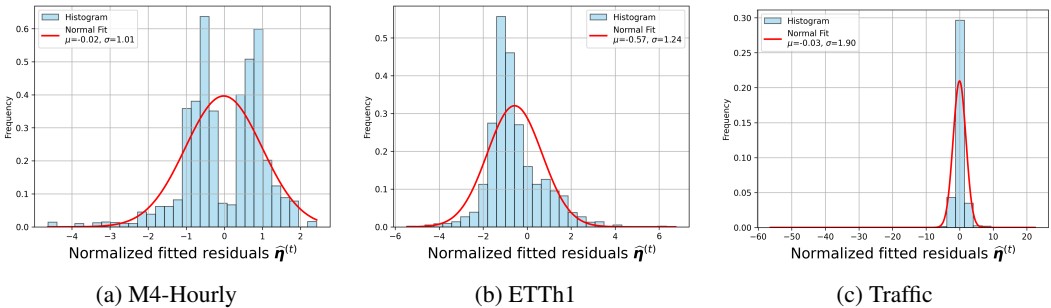

(a) M4-Hourly  (b) ETTh1  (c) Traffic

Figure 2: Histograms of the normalized fitted residuals $\widehat{\eta}_t$ across various datasets. The red lines here represent the Gaussian density curves based on the mean and standard deviation of $\widehat{\eta}_t$.

forecasting, Section B.3 of the Appendix evaluates the mean forecasting performance of various algorithms with and without adding DualRes. All experimental results are based on five repetitions, and we demonstrate the 95% confidence intervals apart from the average metrics.

**Results of univariate probabilistic forecasting.** The performance of DualRes is evaluated through ablation studies in Table 1, where the baseline models are *DeepAR Salinas et al. (2020), DLinear Zeng et al. (2023), PatchTST Nie et al. (2023), and TimeMixer Wang et al. (2024a)*. DLinear, PatchTST, and TimeMixer were originally developed for mean forecasting, and their distributional indices are obtained through fitting a t-distribution to the predictive values, which is the default operation in probabilistic forecasting frameworks such as Alexandrov et al. (2020).

As demonstrated in Table 1, incorporating information on conditional volatility and the distribution of normalized residuals leads to substantial improvements in both CRPS and MAEC across forecasting algorithms—for example, the average CRPS of TimeMixer on the Exchange dataset decreases from 0.027 to 0.014 after applying DualRes. In addition, DualRes enhances the stability of forecasting algorithms, as reflected in achieving narrower confidence intervals.

the CRPS and MAEC of various forecasting algorithms have significant decreases after incorporating information of conditional volatility and the distribution of normalized residuals in forecasting—for example, the average CRPS of TimeMixer when applied to Exchange data decreases from 0.027 to 0.014. Furthermore, DualRes increases the stability of the prediction algorithms in the sense of reaching narrow confidence intervals.

We attribute the performance improvement to DualRes's ability to capture information about both heterogeneity and the normalized residuals distribution. As shown in Figure 3, the widths of the prediction intervals, which are controlled by conditional volatility, vary substantially across different

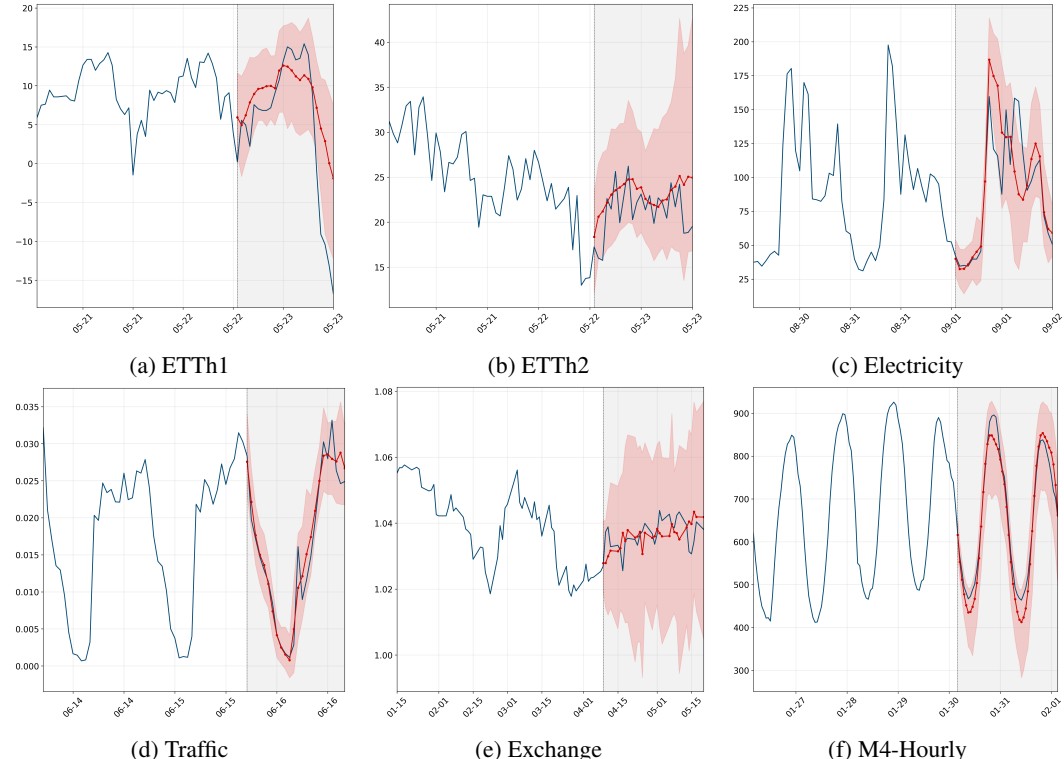

(a) ETTh1      (b) ETTh2      (c) Electricity

(d) Traffic      (e) Exchange      (f) M4-Hourly

Figure 3: Prediction intervals generated by predictive algorithms incorporating DualRes. Blue lines, red lines, and red shadow areas respectively represent the true values, the predictive means, and the 90% prediction intervals.

prediction steps. By explicitly accounting for the volatility, DualRes enhances the performance of forecasting algorithms.

In addition to volatility, Figure 2 shows that the distribution of normalized fitted residuals rarely follows a parametric family, such as the normal or $t$-distribution, in real-world datasets. In practice, these distributions may exhibit multimodality or heavy tails. DualRes avoids the need to impose a parametric assumption—such as those in Zheng et al. (2025)—by introducing a resampling step (Line 3 of Algorithm 2). This design also contributes to its performance gains.

### 5.2 MULTIVARIATE PROBABILISTIC FORECASTING

**Dataset and experimental settings.** We conduct experiments on three real-world datasets: *ETTh1, ETTh2, Electricity,* with a detailed introduction in Section B.1 of the Appendix.

Compared to univariate time series forecasting, multivariate time series data can exhibit spatial dependence, making probabilistic forecasting algorithms essential for capturing spatial dependence. Accordingly, in addition to CRPS and MAEC, we also evaluate the performance of probabilistic forecasting algorithms using the energy score (ES) Chung et al. (2024), with further details provided in Section B.2 of the Appendix.

**Results of multivariate probabilistic forecasting.** The performance of DualRes is evaluated through ablation studies in Table 2, using baseline models *VEC-LSTM* Salinas et al. (2019) and *TMDM* Li et al. (2024). VEC-LSTM, also known as the DeepVAR model, is an RNN-based time series model with a Gaussian copula process output. TMDM is a Transformer-based diffusion model. Both algorithms were originally developed for probabilistic forecasting of multivariate time series.

According to Table 2, DualRes achieves improvements across all metrics for VEC-LSTM and for the majority of metrics in TMDM. For example, on the *Electricity* dataset, the CRPS of TMDM decreases from 0.655 to 0.292 after incorporating DualRes. Apart from accounting for conditional

Table 2: Numerical experiment results on multivariate time series datasets. The interpretation of the values and the use of boldface are the same as in Table 1.

| Dataset | ETTh1 | | | ETTh2 | | | Electricity | | |
|---|---|---|---|---|---|---|---|---|---|
| Metrics | CRPS | MAEC | ES | CRPS | MAEC | ES | CRPS | MAEC | ES |
| VEC-LSTM | 0.184(0.003) | 0.310(0.015) | 3.873(0.157) | 0.095(0.002) | 0.243(0.014) | 6.423(0.196) | 0.441(0.014) | 0.385(0.072) | 48684(3323) |
| +Ours | **0.182(0.005)** | **0.294(0.001)** | **3.503(0.085)** | **0.087(0.001)** | **0.241(0.016)** | **6.067(0.190)** | **0.301(0.013)** | **0.251(0.009)** | **41398(3744)** |
| TMDM | 0.456(0.023) | **0.268(0.052)** | 13.344(0.163) | 0.092(0.008) | 0.318(0.123) | **6.933(0.393)** | 0.655(0.275) | 0.458(0.082) | 87761(6179) |
| +Ours | **0.397(0.040)** | 0.458(0.082) | **11.341(0.372)** | **0.092(0.004)** | **0.306(0.023)** | 7.326(0.498) | **0.292(0.018)** | **0.227(0.009)** | **37322(2438)** |

heteroskedasticity and residual distributional information, the improvement in the energy score highlights DualRess ability to capture spatial dependence in multivariate time series. This effectiveness stems from resampling entire normalized residual vectors $\widehat{\boldsymbol{\eta}}_t$, rather than their individual components.

## 6 DISCUSSION

Focusing on probabilistic time series forecasting, this manuscript proposes the DualRes framework, which extracts conditional volatility information from fitted residuals and models the distribution of normalized residuals through resampling. These operations make DualRes robust to conditional heteroskedasticity and free from restrictive parametric assumptions, such as Gaussianity. We further provide theoretical guarantees for the validity of the proposed training and inference procedures.

In addition, as DualRes requires only conditional mean forecasts, it offers substantial flexibility in the choice of models for both the conditional mean and volatility. As demonstrated in the numerical experiments, even models originally designed for mean forecasting can be adapted for probabilistic forecasting, leading to significant performance gains.

Our work highlights the importance of incorporating the distribution of normalized residuals—beyond conditional mean and volatility—in probabilistic forecasting. Since residuals in real-world time series often deviate from parametric distributions, introducing a resampling step enables greater flexibility when addressing the underlying randomness in the data.

**Limitations and Future Work.** One main limitation of our work lies in the computational complexity of the algorithm. Concerning this, one potential future direction of this work involves leveraging advanced subsampling techniques, like those in McElroy & Politis (2024), to decrease computational complexity.

Another limitation is that the validity of Theorem 1 depends on the conditional mean and volatility models accurately reflecting the true conditional mean and volatility functions. As a result, if future observations have a distributional shift, the proposed method may no longer be reliable.

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

## A   PROOF OF THEOREM 1

To validate Theorem 1, we propose the following technical assumptions.

**Assumptions:**

1. $\boldsymbol{\eta}_t, t = 1, 2, \cdots$, are independent and identically distributed with continuous cumulative distribution function $P(\cdot) : \mathbb{R}^d \to \mathbb{R}$. Suppose $\mathbf{E}\left[\boldsymbol{\eta}_1\right] = 0$ and $\mathrm{Var}(\boldsymbol{\eta}_{1,i}) \leq C$ for a constant $C$ and any $i = 1, \cdots, d$.

2. For a vector $\boldsymbol{x} \in \mathbf{R}^d$, define $||\boldsymbol{x}||$ as its $L^2$ norm. We suppose the conditional mean and volatility function estimator satisfy

$$\sup_{\mathbf{Y} \in \mathbf{R}^{d \times q}} ||\widehat{F}(\mathbf{Y}) - F(\mathbf{Y})|| \to_p 0 \quad \text{and} \quad \sup_{\mathbf{Y} \in \mathbf{R}^{d \times s}} |\widehat{G}_i(\mathbf{Y}) - G_i(\mathbf{Y})| \to_p 0,$$

where $i = 1, 2, \cdots, d$, and $\to_p$ denotes convergence in probability.

3. Suppose $G_i(\cdot)$ is continuous differentiable with bounded gradient, i.e.,

$$\sup_{\mathbf{Y} \in \mathbf{R}^{d \times s}} ||\nabla_{\mathbf{Y}} G_i(\mathbf{Y})|| < \infty$$

for $i = 1, \cdots, d$. Furthermore, suppose there exists a constant $c > 0$ such that

$$\inf_{\mathbf{Y} \in \mathbf{R}^{d \times s}} |G_i(\mathbf{Y})| > c$$

for $i = 1, \cdots, d$.

With those assumptions, we demonstrate that Theorem 1 holds true.

*Proof of Theorem 1.* For any vector $\mathbf{y} = (\mathbf{y}_1, \cdots, \mathbf{y}_d)^\top \in \mathbf{R}^d$, define

$$\widetilde{P}(\boldsymbol{y}) = \frac{1}{T - q - s} \sum_{t=s+q+1}^{T} \mathbf{1}_{\boldsymbol{\eta}_t \leq \boldsymbol{y}}.$$

From Glivenko-Cantelli Theorem, like Theorem 4 of Sharipov (2011), we have

$$\sup_{\boldsymbol{y} \in \mathbf{R}^d} |\widetilde{G}(\boldsymbol{y}) - G(\boldsymbol{y})| \to_p 0.$$

On the other hand, define the functions

$$g_0(u) = (1 - \min(1, \max(u, 0))^4)^4 \quad \text{and} \quad g_{\psi,t}(x) = g_0(\psi(x - t)),$$

as demonstrated in Xu et al. (2019), which satisfy the following property: $g_0(\cdot)$ is third-order continuous differentiable, $g_0(u) = 1$ if $u \leq 0$, $g_0(u) = 0$ if $u \geq 1$, and

$$g_* = \sup_{u \in \mathbf{R}} \{|g_0'(u)| + |g_0''(u)| + |g_0'''(u)|\} < \infty, \ \mathbf{1}_{x \leq t} \leq g_{\psi,t}(x) \leq \mathbf{1}_{x \leq t+\psi^{-1}}, \ \sup_{x,t \in \mathbf{R}} |g_{\psi,t}'(x)| \leq g_*\psi.$$

Define

$$\begin{aligned}
\boldsymbol{\Delta}_t &= \widehat{\boldsymbol{\eta}}_t - \boldsymbol{\eta}_t \\
&= \widehat{G}^{-1}\left(\widehat{\boldsymbol{\zeta}}_{t-s}, \cdots \widehat{\boldsymbol{\zeta}}_{t-1}\right)\left(F(\mathbf{x}_{t-q}, \cdots, \mathbf{x}_{t-1}) - \widehat{F}(\mathbf{x}_{t-q}, \cdots, \mathbf{x}_{t-1})\right) \\
&\quad + \widehat{G}^{-1}\left(\widehat{\boldsymbol{\zeta}}_{t-s}, \cdots \widehat{\boldsymbol{\zeta}}_{t-1}\right)\left(G\left(\boldsymbol{\zeta}_{t-s}, \cdots \boldsymbol{\zeta}_{t-1}\right) - \widehat{G}\left(\widehat{\boldsymbol{\zeta}}_{t-s}, \cdots \widehat{\boldsymbol{\zeta}}_{t-1}\right)\right) \boldsymbol{\eta}_t.
\end{aligned}$$

Notice that

$$\widehat{F}(\boldsymbol{y}) = \frac{1}{T - q - s} \sum_{t=s+q+1}^{T} \mathbf{1}_{\boldsymbol{\eta}_t + \boldsymbol{\Delta}_t \leq \boldsymbol{y}} \leq \frac{1}{T - q - s} \sum_{t=s+q+1}^{T} \prod_{i=1}^{d} g_{\psi,\boldsymbol{y}_i}(\boldsymbol{\eta}_{t,i} + \boldsymbol{\Delta}_{t,i}).$$

From Taylor expansion,

$$|\prod_{i=1}^{d} g_{\psi,\boldsymbol{y}_i}(\boldsymbol{\eta}_{t,i} + \boldsymbol{\Delta}_{t,i}) - \prod_{i=1}^{d} g_{\psi,\boldsymbol{y}_i}(\boldsymbol{\eta}_{t,i})|$$

$$\leq \sum_{i=1}^{d} (\prod_{j=1}^{i-1} g_{\psi,\boldsymbol{y}_i}(\boldsymbol{\eta}_{t,i} + \boldsymbol{\Delta}_{t,i})(g_{\psi,\boldsymbol{y}_i}(\boldsymbol{\eta}_{t,i} + \boldsymbol{\Delta}_{t,i}) - g_{\psi,\boldsymbol{y}_i}(\boldsymbol{\eta}_{t,i})) \prod_{j=i+1}^{d} g_{\psi,\boldsymbol{y}_i}(\boldsymbol{\eta}_{t,i})$$

$$\leq \sum_{i=1}^{d} |g_{\psi,\boldsymbol{y}_i}(\boldsymbol{\eta}_{t,i} + \boldsymbol{\Delta}_{t,i}) - g_{\psi,\boldsymbol{y}_i}(\boldsymbol{\eta}_{t,i})| \leq g_* \psi \sum_{i=1}^{d} |\boldsymbol{\Delta}_{t,i}| \leq g_* \psi \sqrt{d} ||\boldsymbol{\Delta}_t||.$$

Therefore,

$$\frac{1}{T-q-s} \sum_{t=s+q+1}^{T} \prod_{i=1}^{d} g_{\psi,\boldsymbol{y}_i}(\boldsymbol{\eta}_{t,i} + \boldsymbol{\Delta}_{t,i})$$

$$\leq \frac{1}{T-q-s} \sum_{t=s+q+1}^{T} \prod_{i=1}^{d} g_{\psi,\boldsymbol{y}_i}(\boldsymbol{\eta}_{t,i}) + \frac{g_* \psi \sqrt{d}}{T-q-s} \sum_{t=s+q+1}^{T} ||\boldsymbol{\Delta}_t||$$

$$\leq \frac{1}{T-q-s} \sum_{t=s+q+1}^{T} \mathbf{1}_{\boldsymbol{\eta}_t \leq \boldsymbol{y}+\psi^{-1}} + \frac{g_* \psi \sqrt{d}}{T-q-s} \sum_{t=s+q+1}^{T} ||\boldsymbol{\Delta}_t||$$

$$= \widetilde{F}(\boldsymbol{y} + \psi^{-1}\mathbf{h}) + \frac{g_* \psi \sqrt{d}}{T-q-s} \sum_{t=s+q+1}^{T} ||\boldsymbol{\Delta}_t||,$$

where $\mathbf{h} = (1, 1, \cdots, 1)^\top$. Similarly,

$$\widehat{F}(\boldsymbol{y}) \geq \frac{1}{T-q-s} \sum_{t=s+q+1}^{T} \prod_{i=1}^{d} g_{\psi,\boldsymbol{y}_i-\psi^{-1}}(\boldsymbol{\eta}_{t,i} + \boldsymbol{\Delta}_{t,i})$$

$$\geq \frac{1}{T-q-s} \sum_{t=s+q+1}^{T} \prod_{i=1}^{d} g_{\psi,\boldsymbol{y}_i-\psi^{-1}}(\boldsymbol{\eta}_{t,i}) - \frac{g_* \psi \sqrt{d}}{T-q-s} \sum_{t=s+q+1}^{T} ||\boldsymbol{\Delta}_t||$$

$$\geq \widetilde{F}(\boldsymbol{y} - \psi^{-1}\mathbf{h}) - \frac{g_* \psi \sqrt{d}}{T-q-s} \sum_{t=s+q+1}^{T} ||\boldsymbol{\Delta}_t||.$$

With probability tending to 1,

$$\inf_{\mathbf{Y} \in \mathbf{R}^{d \times s}} \widehat{G}_i(\mathbf{Y}) \geq \inf_{\mathbf{Y} \in \mathbf{R}^{d \times s}} G_i(\mathbf{Y}) - \sup_{\mathbf{Y} \in \mathbf{R}^{d \times s}} |\widehat{G}_i(\mathbf{Y}) - G_i(\mathbf{Y})| > c/2.$$

If that happens for $i = 1, \cdots, d$, we have

$$||\widehat{G}^{-1} \left( \widehat{\boldsymbol{\zeta}}_{t-s}, \cdots \widehat{\boldsymbol{\zeta}}_{t-1} \right) \left( F(\mathbf{x}_{t-q}, \cdots, \mathbf{x}_{t-1}) - \widehat{F}(\mathbf{x}_{t-q}, \cdots, \mathbf{x}_{t-1}) \right) ||$$

$$\leq \frac{2}{c} \sup_{\mathbf{Y} \in \mathbf{R}^{d \times q}} ||F(\mathbf{Y}) - \widehat{F}(\mathbf{Y})|| \rightarrow_p 0. \tag{9}$$

On the other hand, for any $i = 1, \cdots, d$, the $i$th element of $\widehat{G}^{-1} \left( \widehat{\boldsymbol{\zeta}}_{t-s}, \cdots \widehat{\boldsymbol{\zeta}}_{t-1} \right) \left( G \left( \boldsymbol{\zeta}_{t-s}, \cdots \boldsymbol{\zeta}_{t-1} \right) - \widehat{G} \left( \widehat{\boldsymbol{\zeta}}_{t-s}, \cdots \widehat{\boldsymbol{\zeta}}_{t-1} \right) \right) \boldsymbol{\eta}_t$ is

$$\frac{G_i \left( \boldsymbol{\zeta}_{t-s}, \cdots \boldsymbol{\zeta}_{t-1} \right) - \widehat{G}_i \left( \widehat{\boldsymbol{\zeta}}_{t-s}, \cdots \widehat{\boldsymbol{\zeta}}_{t-1} \right)}{\widehat{G}_i \left( \widehat{\boldsymbol{\zeta}}_{t-s}, \cdots \widehat{\boldsymbol{\zeta}}_{t-1} \right)} \boldsymbol{\eta}_{t,i}.$$

and

$$
|\frac{G_i\left(\boldsymbol{\zeta}_{t-s}, \cdots \boldsymbol{\zeta}_{t-1}\right) - \widehat{G}_i\left(\widehat{\boldsymbol{\zeta}}_{t-s}, \cdots \widehat{\boldsymbol{\zeta}}_{t-1}\right)}{\widehat{G}_i\left(\widehat{\boldsymbol{\zeta}}_{t-s}, \cdots \widehat{\boldsymbol{\zeta}}_{t-1}\right)}\boldsymbol{\eta}_{t,i}|
$$

$$
\leq \frac{2|\boldsymbol{\eta}_{t,i}|}{c}\left(|G_i\left(\boldsymbol{\zeta}_{t-s}, \cdots \boldsymbol{\zeta}_{t-1}\right) - G_i\left(\widehat{\boldsymbol{\zeta}}_{t-s}, \cdots \widehat{\boldsymbol{\zeta}}_{t-1}\right)|\right.
$$

$$
\left. + |G_i\left(\widehat{\boldsymbol{\zeta}}_{t-s}, \cdots \widehat{\boldsymbol{\zeta}}_{t-1}\right) - \widehat{G}_i\left(\widehat{\boldsymbol{\zeta}}_{t-s}, \cdots \widehat{\boldsymbol{\zeta}}_{t-1}\right)|\right)
$$

From Assumption 2,

$$
|G_i\left(\widehat{\boldsymbol{\zeta}}_{t-s}, \cdots \widehat{\boldsymbol{\zeta}}_{t-1}\right) - \widehat{G}_i\left(\widehat{\boldsymbol{\zeta}}_{t-s}, \cdots \widehat{\boldsymbol{\zeta}}_{t-1}\right)| \leq \sup_{\mathbf{Y}\in\mathbf{R}^{d\times s}}|G_i\left(\mathbf{Y}\right) - \widehat{G}_i\left(\mathbf{Y}\right)| \to_p 0. \quad (10)
$$

On the other hand, for any $t = q+1, \cdots, T$,

$$
||\widehat{\boldsymbol{\zeta}}_t - \boldsymbol{\zeta}_t|| = ||F(\mathbf{x}_{t-q}, \cdots, \mathbf{x}_{t-1}) - \widehat{F}(\mathbf{x}_{t-q}, \cdots, \mathbf{x}_{t-1})||
$$

$$
\leq \sup_{\mathbf{Y}\in\mathbf{R}^{d\times q}}||F(\mathbf{Y}) - \widehat{F}(\mathbf{Y})|| \to_p 0.
$$

Define the matrix

$$
\boldsymbol{\Gamma} = \begin{bmatrix} \widehat{\boldsymbol{\zeta}}_{t-s} - \boldsymbol{\zeta}_{t-s} & \cdots & \widehat{\boldsymbol{\zeta}}_{t-1} - \boldsymbol{\zeta}_{t-1} \end{bmatrix},
$$

from Taylor's expansion,

$$
|G_i\left(\boldsymbol{\zeta}_{t-s}, \cdots \boldsymbol{\zeta}_{t-1}\right) - G_i\left(\widehat{\boldsymbol{\zeta}}_{t-s}, \cdots \widehat{\boldsymbol{\zeta}}_{t-1}\right)| = |\sum_{i=1}^{d}\sum_{j=1}^{s}(\nabla_{\mathbf{Z}}G_i(\mathbf{Z}))_{ij}\boldsymbol{\Gamma}_{ij}|
$$

$$
\leq \sum_{i=1}^{d}\sum_{j=1}^{s}|\nabla_{\mathbf{Z}}G_i(\mathbf{Z}))_{ij}||\boldsymbol{\Gamma}_{ij}| \quad (11)
$$

$$
\leq Cds\sup_{\mathbf{Y}\in\mathbf{R}^{d\times q}}||F(\mathbf{Y}) - \widehat{F}(\mathbf{Y})||,
$$

where $\mathbf{Z} \in \mathbf{R}^{d\times s}$ is a random matrix. From eq.equation 9, eq.equation 10 and eq.equation 11, with probability tending to 1

$$
||\boldsymbol{\Delta}_t|| \leq \frac{2}{c}\sup_{\mathbf{Y}\in\mathbf{R}^{d\times q}}||F(\mathbf{Y}) - \widehat{F}(\mathbf{Y})|| + \sqrt{\sum_{i=1}^{d}\left(\frac{G_i\left(\boldsymbol{\zeta}_{t-s}, \cdots \boldsymbol{\zeta}_{t-1}\right) - \widehat{G}_i\left(\widehat{\boldsymbol{\zeta}}_{t-s}, \cdots \widehat{\boldsymbol{\zeta}}_{t-1}\right)}{\widehat{G}_i\left(\widehat{\boldsymbol{\zeta}}_{t-s}, \cdots \widehat{\boldsymbol{\zeta}}_{t-1}\right)}\boldsymbol{\eta}_{t,i}\right)^2}
$$

$$
\leq \frac{2}{c}\sup_{\mathbf{Y}\in\mathbf{R}^{d\times q}}||F(\mathbf{Y}) - \widehat{F}(\mathbf{Y})|| + \frac{2\sqrt{d}}{c}\max_{i=1,\cdots,d}|\boldsymbol{\eta}_{t,i}| \times |G_i\left(\boldsymbol{\zeta}_{t-s}, \cdots \boldsymbol{\zeta}_{t-1}\right) - \widehat{G}_i\left(\widehat{\boldsymbol{\zeta}}_{t-s}, \cdots \widehat{\boldsymbol{\zeta}}_{t-1}\right)|
$$

$$
\leq \frac{2}{c}\sup_{\mathbf{Y}\in\mathbf{R}^{d\times q}}||F(\mathbf{Y}) - \widehat{F}(\mathbf{Y})|| + \frac{2\sqrt{d}}{c}\left(\sum_{i=1}^{d}|\boldsymbol{\eta}_{t,i}|\right)\left(\sup_{\mathbf{Y}\in\mathbf{R}^{d\times s}}|G_i\left(\mathbf{Y}\right) - \widehat{G}_i\left(\mathbf{Y}\right)|\right)
$$

$$
+ \frac{2\sqrt{d}}{c}\left(\sum_{i=1}^{d}|\boldsymbol{\eta}_{t,i}|\right)\left(Cds\sup_{\mathbf{Y}\in\mathbf{R}^{d\times q}}||F(\mathbf{Y}) - \widehat{F}(\mathbf{Y})||\right).
$$

Since

$$\frac{\psi\sqrt{d}}{T-q-s}\sum_{t=s+q+1}^{T}||\boldsymbol{\Delta}_t|| \leq \frac{2\psi\sqrt{d}}{c}\sup_{\mathbf{Y}\in\mathbf{R}^{d\times q}}||F(\mathbf{Y})-\widehat{F}(\mathbf{Y})||$$

$$+ \frac{2\psi d}{c(T-q-s)}\sum_{i=1}^{d}\sup_{\mathbf{Y}\in\mathbf{R}^{d\times s}}|G_i(\mathbf{Y})-\widehat{G}_i(\mathbf{Y})|\sum_{t=s+q+1}^{T}|\boldsymbol{\eta}_{t,i}|$$

$$+ \frac{2C\psi d^2 s}{c(T-q-s)}\sup_{\mathbf{Y}\in\mathbf{R}^{d\times q}}||F(\mathbf{Y})-\widehat{F}(\mathbf{Y})||\sum_{i=1}^{d}\sum_{t=s+q+1}^{T}|\boldsymbol{\eta}_{t,i}|$$

$$\leq \frac{2\psi\sqrt{d}}{c}\sup_{\mathbf{Y}\in\mathbf{R}^{d\times q}}||F(\mathbf{Y})-\widehat{F}(\mathbf{Y})||$$

$$+ \frac{2\psi d}{c(T-q-s)}\left(\max_{i=1,\cdots,d}\sup_{\mathbf{Y}\in\mathbf{R}^{d\times s}}|G_i(\mathbf{Y})-\widehat{G}_i(\mathbf{Y})|\right)\left(\sum_{i=1}^{d}\sum_{t=s+q+1}^{T}|\boldsymbol{\eta}_{t,i}|\right)$$

$$+ \frac{2C\psi d^2 s}{c(T-q-s)}\sup_{\mathbf{Y}\in\mathbf{R}^{d\times q}}||F(\mathbf{Y})-\widehat{F}(\mathbf{Y})||\sum_{i=1}^{d}\sum_{t=s+q+1}^{T}|\boldsymbol{\eta}_{t,i}|,$$

and

$$\mathbf{E}\left[\frac{1}{T-q-s}\sum_{i=1}^{d}\sum_{t=s+q+1}^{T}|\boldsymbol{\eta}_{t,i}|\right] = \sum_{i=1}^{d}\mathbf{E}\left[|\boldsymbol{\eta}_{1,i}|\right] < \infty.$$

According to Assumption 2,

$$\frac{\psi\sqrt{d}}{T-q-s}\sum_{t=s+q+1}^{T}||\boldsymbol{\Delta}_t|| \to_p 0,$$

and the result is proven according to the continuity of $P(\cdot)$, and by setting $\psi \to \infty$. $\qquad\square$

# B  ADDITIONAL EXPERIMENTAL RESULTS

## B.1  INTRODUCTION OF DATASETS AND HYPER-PARAMETERS

Our work evaluates the performance of models on six commonly used datasets named *ETTh1, ETTh2, Electricity, Traffic, Exchange, M4-Hourly* when performing univariate probabilistic forecasting, and on three datasets *ETTh1, ETTh2, Electricity* when performing multivariate probabilistic forecasting. The names and characteristics of the datasets are summarized as in Table 3. *Electricity, Traffic, Exchange, M4-Hourly* are available in GluonTS Alexandrov et al. (2020). We consider the *ETTh1, ETTh2, Electricity* datasets as multiple separate univariate time series in univariate experiments, while we consider them as single multivariate time series data in multivariate experiments.

Table 3: Overview of the datasets used in univariate time series experiments.

| Dataset | GluonTS Name | Dimension | Test | Domain | Freq. | Median Time Steps |
|---|---|---|---|---|---|---|
| ETTh1[1] | - | 7 | 126 | $\mathbf{R}^+$ | H | 17396 |
| ETTh2[2] | - | 7 | 126 | $\mathbf{R}^+$ | H | 17396 |
| M4-Hourly[3] | m4_hourly | 414 | 414 | $\mathbb{N}$ | H | 960 |
| Electricity[4] | electricity_nips | 370 | 2590 | $\mathbf{R}^+$ | H | 5833 |
| Traffic[5] | traffic_nips | 963 | 6741 | $(0,1)$ | H | 4001 |
| Exchange[6] | exchange_rate_nips | 8 | 40 | $\mathbf{R}^+$ | D | 6071 |

---

[1] https://github.com/zhouhaoyi/ETDataset/tree/main
[2] https://github.com/zhouhaoyi/ETDataset/tree/main
[3] https://github.com/Mcompetitions/M4-methods/tree/master/Dataset
[4] ttps://archive.ics.uci.edu/dataset/321/electricityloaddiagrams20112014
[5] https://zenodo.org/records/4656132
[6] https://github.com/laiguokun/multivariate-time-series-data

For the experiment detail, we set the resample times 100 when computing the CRPS and MAEC metrics. The context length and prediction length in conditional mean model follow the settings in Kollovieh et al. (2023). In our work, for univariate time series data, we use the technique mentioned in Remarks 1 and 2, and adopt a simple multilayer perceptron model (referred to as "SimpleFeedForwardEstimator in the *GluonTS* package Alexandrov et al. (2020)) to model the logarithm of the conditional volatilities. For multivariate time series, we use the *VEC-LSTM* model to estimate the logarithm of conditional volatilities in the first experiment, and the *TMDM* model in the second one. The context length of the conditional volatility model is selected based on the autocorrelation coefficients plot (Figure 4) below. The prediction length in the conditional volatility model is set to 1. All other hyperparameters are set to their default values in the GluonTS package.

Table 4: Hyperparameters of the Conditional Mean and Volatility model

| Dataset | Conditional Mean Model | | Conditional Volatility Model | |
| --- | --- | --- | --- | --- |
| | Context Len. | Predict Len. | Context Len. | Predict Len. |
| ETTh1 | 336 | 24 | 24 | 1 |
| ETTh2 | 336 | 24 | 24 | 1 |
| M4-Hourly | 312 | 48 | 14 | 1 |
| Electricity | 336 | 24 | 48 | 1 |
| Traffic | 336 | 24 | 48 | 1 |
| Exchange | 360 | 30 | 100 | 1 |

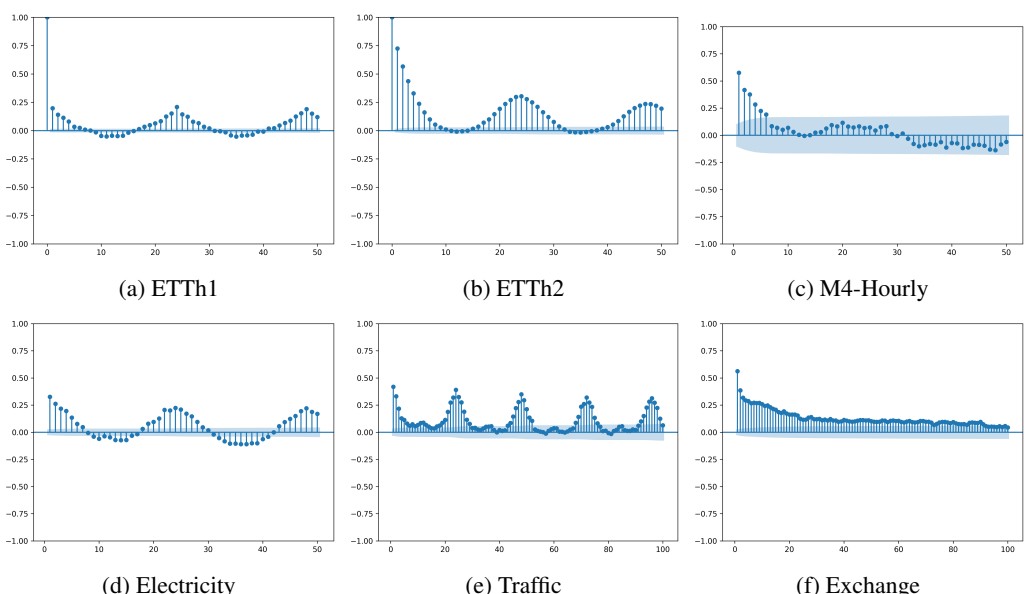

(a) ETTh1      (b) ETTh2      (c) M4-Hourly

(d) Electricity      (e) Traffic      (f) Exchange

Figure 4: Autocorrelation coefficients plot of the logarithm of square fitted residuals.

## B.2 METRICS OF THE EXPERIMENT

**Continuous Ranked Probability Score (CRPS).** The CRPS is a commonly used metric in probabilistic forecasting, as demonstrated in Gneiting & Raftery (2007) and Kollovieh et al. (2023). It is defined as the integral of the pinball loss over the interval $[0, 1]$:

$$CRPS(F^{-1}, y) = \int_0^1 2\Lambda_\kappa(F^{-1}(\kappa), y)\mathrm{d}\kappa, \text{ where } \Lambda_\kappa(q, y) = (\kappa - \mathbf{1}_{y<q}) \times (y - q).$$

A forecasted quantile function $F^{-1}$ with a small CRPS indicates good alignment with the observation $y$. We approximate the quantile function by sample quantiles at nine quantile levels $\{10\%, 20\%, \cdots, 90\%\}$. These sample quantiles are estimated from 100 forecast samples.

For multivariate time series, the CRPS is computed as the summation of the element-wise CRPS.

**Mean Absolute Error of Coverage (MAEC).** Suppose the prediction step is $J$, and the prediction intervals are with endpoints $\mathbf{u}_j, \mathbf{v}_j \in \mathbf{R}^d$, where $\mathbf{u}_{j,i} \leq \mathbf{v}_{j,i}$ for $i = 1, \cdots, d$, here $j = 1, \cdots, J$. The coverage probability we are interested in is the frequency

$$\widehat{p}(\beta) = \frac{1}{dJ} \sum_{j=1}^{J} \sum_{i=1}^{d} \mathbf{1}_{\mathbf{u}_{j,i} \leq \mathbf{x}_{T+j,i} \leq \mathbf{v}_{j,i}},$$

$\beta$ here indicates the quantile level of the prediction intervals. Specifically, for univariate time series ($d = 1$), the endpoints of prediction intervals are scalars, and the coverage probability becomes

$$\widehat{p}(\beta) = \frac{1}{J} \sum_{j=1}^{J} \mathbf{1}_{\mathbf{u}_{j,1} \leq \mathbf{x}_{T+j} \leq \mathbf{v}_{j,1}}.$$

We consider 9 quantile levels $\{\beta_1, \cdots, \beta_9\} = 10\%, 20\%, \cdots, 90\%$, and the MAEC metric calculates the mean absolute error between $\widehat{p}(\beta_s)$ and $\beta_s$, i.e.,

$$MAEC = \sum_{s=1}^{9} |\widehat{p}(\beta_s) - \beta_s|.$$

A low MAEC indicates that the prediction intervals achieve the desired coverage probabilities in general, thereby reflecting higher accuracy of prediction intervals.

**Energy Score (ES).** Introduced in Chung et al. (2024), ES is a metric to evaluate the performance of a probabilistic forecasting method in capturing spatial dependence for multivariate data. For a future time series data $\mathbf{y}_j \in \mathbf{R}^d$, and a predictive distribution $\widehat{p}_j$, we define the energy score as

$$ES_j = \mathbb{E}_{\mathbf{x} \sim \widehat{p}_j} ||\mathbf{x} - \mathbf{y}_j||_2^\beta - \frac{1}{2} \mathbb{E}_{\mathbf{x}, \mathbf{x}' \sim \widehat{p}_j} ||\mathbf{x} - \mathbf{x}'||_2^\beta,$$

where $\mathbf{x}, \mathbf{x}'$ are independent sampled from $\widehat{p}_j$. We calculate the ES as the average value

$$ES = \frac{1}{J} \sum_{j=1}^{J} ES_j.$$

Following Chung et al. (2024), we set $\beta = 1.7$. A smaller energy score indicates that the predictive distribution is closer to the ground truth.

In addition to the probabilistic forecasting metrics, we evaluate the mean forecasting performance of univariate time series through the metrics *Normalized Deviation (ND)* and *normalized root mean squared error (NRMSE),* introduced as follows:

**Normalized Deviation (ND).** Suppose the future $J$ observations are $\mathbf{x}_{T+1}, \cdots, \mathbf{x}_{T+J}$ with corresponding predictors $\widehat{\mathbf{x}}_{T+j}$, ND is defined by

$$ND = \frac{\sum_{j=1}^{J} |\widehat{\mathbf{x}}_{T+j} - \mathbf{x}_{T+j}|}{\sum_{j=1}^{J} |\mathbf{x}_{T+j}|},$$

indicating the absolute error normalized by the total absolute scale of the prediction time series. ND is independent of the scale of the time series, making it suitable for comparison across different datasets.

**Normalized root mean squared error (NRMSE).** With the notations in ND, the NRMSE is defined by

$$\frac{RMSE}{|\mathbf{x}|}, \quad \text{where} \quad RMSE = \sqrt{\frac{1}{J} \sum_{j=1}^{J} (\widehat{\mathbf{x}}_{T+j} - \mathbf{x}_{T+j})^2} \quad \text{and} \quad \overline{|\mathbf{x}|} = \frac{1}{J} \sum_{j=1}^{J} |\mathbf{x}_{T+j}|.$$

Similar to ND, NRMSE is also independent of the scale of time series.

Table 5: Mean forecasting performance. The interpretation of the values and the use of boldface are the same as in Table 1.

| Models | Metrics | ETTh1 | ETTh2 | Electricity | Traffic | Exchange | M4-Hourly |
|--------|---------|-------|-------|-------------|---------|----------|-----------|
| DeepAR | ND | 0.225(0.045) | **0.082(0.011)** | 0.104(0.001) | **0.128(0.012)** | 0.019(0.002) | 0.109(0.113) |
|  | NRMSE | 0.417(0.063) | 0.123(0.015) | 0.760(0.010) | **0.391(0.052)** | 0.029(0.002) | 0.653(0.515) |
| DeepAR +Ours | ND | **0.219(0.018)** | 0.114(0.017) | **0.086(0.002)** | 0.154(0.010) | **0.013(0.001)** | **0.054(0.002)** |
|  | NRMSE | **0.408(0.026)** | **0.092(0.091)** | **0.625(0.066)** | 0.429(0.037) | **0.020(0.001)** | **0.296(0.017)** |
| DLinear | ND | **0.227(0.004)** | 0.086(0.011) | 0.075(0.009) | 0.161(0.002) | 0.024(0.010) | 0.057(0.006) |
|  | NRMSE | **0.422(0.004)** | 0.126(0.012) | 0.593(0.063) | **0.407(0.002)** | 0.044(0.026) | 0.323(0.050) |
| DLinear +Ours | ND | 0.243(0.011) | **0.086(0.007)** | **0.067(0.000)** | **0.160(0.004)** | **0.012(0.002)** | **0.046(0.015)** |
|  | NRMSE | 0.452(0.016) | **0.097(0.090)** | **0.538(0.017)** | 0.418(0.001) | **0.020(0.003)** | **0.315(0.093)** |
| PatchTST | ND | **0.212(0.003)** | **0.084(0.017)** | **0.078(0.004)** | **0.151(0.002)** | 0.017(0.005) | **0.053(0.011)** |
|  | NRMSE | **0.402(0.001)** | 0.122(0.021) | **0.635(0.004)** | 0.441(0.003) | 0.024(0.005) | **0.283(0.081)** |
| PatchTST +Ours | ND | 0.247(0.059) | 0.090(0.002) | 0.080(0.003) | 0.159(0.004) | **0.015(0.002)** | 0.063(0.031) |
|  | NRMSE | 0.450(0.095) | **0.081(0.058)** | 0.656(0.058) | **0.437(0.006)** | **0.023(0.004)** | 0.615(0.061) |
| TimeMixer | ND | **0.460(0.005)** | 0.120(0.004) | 0.382(0.011) | **0.498(0.004)** | 0.030(0.014) | **0.142(0.012)** |
|  | NRMSE | **0.855(0.021)** | **0.182(0.009)** | 3.656(0.002) | 0.764(0.003) | 0.041(0.019) | 0.825(0.083) |
| TimeMixer +Ours | ND | 0.461(0.021) | **0.119(0.002)** | **0.379(0.007)** | 0.499(0.001) | **0.015(0.001)** | 0.157(0.007) |
|  | NRMSE | 0.909(0.084) | 0.590(0.530) | **3.599(0.051)** | **0.763(0.003)** | **0.028(0.001)** | **0.605(0.685)** |

## B.3    ADDITIONAL EXPERIMENTAL RESULTS

Table 5 reports the performance of DualRes in mean forecasting, evaluated using the metrics ND and NRMSE. Although the primary goal of DualRes is to improve probabilistic forecasting, the framework also enhances mean forecasting performance and increases the stability of predictive algorithms. We attribute this improvement to the iterative updates in equation 4 of Algorithm 2: since $\widehat{F}$ is a nonlinear function, adding the residuals $\zeta^*_{T+j}$ and applying repeated function compositions alter the distributions—and consequently the means—of the pseudo-samples at future steps.

