# OpenReview forum: "DualRes: A Resampling-Based Framework for Enhancing Probabilistic Forecasting"
_ICLR.cc/2026/Conference — ICLR 2026 Conference Withdrawn Submission_

### Official Review · Reviewer_QAiM · 2025-10-27

**Soundness:** 2
**Presentation:** 2
**Contribution:** 2
**Rating:** 2
**Confidence:** 5

**Summary:**

This paper proposes a post-hoc method for probabilistic forecasting for time series data. The key point of the method is to use two separate models to capture the conditional mean and conditional variance of the target variable, respectively, without assumptions on the distribution of the target variable.

**Strengths:**

The paper addresses an important problem in time series forecasting, i.e., probabilistic forecasting without distributional assumptions, and proposes a novel and a conceptually intuitive method to tackle it.

**Weaknesses:**

- There is a serious omission in the comparison or discussion of conformal prediction methods applied to time series forecasting, which are relevant and strong baselines for the proposed method. Examples are models like HopCPT [1], KOWCPI [2], SPCI [3], and others. These methods provide post-hoc distribution-free uncertainty quantification and should be compared against or at least discussed in relation to the proposed method.
- Multivariate results appear additive rather than integral to the paper contribution. The multivariate extension is not well motivated or explained, also considering the diagonality assumption in the volatility matrix $G$.
- The experimental evaluation is very limited. While the method is evaluated on several datasets, the baselines used for comparison are few and other meaningful metrics should be considered, such as calibration metrics (e.g., coverage probability, interval width) and sharpness.
- The computational efficiency of the proposed method is only mentioned briefly in the conclusion. A more thorough analysis of the computational cost compared to other methods should be provided.
- The presentation of the method and results could be improved for clarity. Some sections are dense and could benefit from more detailed explanations and visualizations to aid understanding.
- There are several minor issues and typos throughout the paper that need to be addressed for better readability, especially in core parts like Sections 3 and 4.
- The code to reproduce the experiments is not provided, which hinders reproducibility and further validation of the results.

## References

[1] Andreas Auer and Martin Gauch and Daniel Klotz and Sepp Hochreiter. Conformal Prediction for Time Series with Modern Hopfield Networks. NeurIPS 2023.

[2] Jonghyeok Lee and Chen Xu and Yao Xie. Kernel-based Optimally Weighted Conformal Prediction Intervals. ICLR 2025.

[3] Chen Xu and Yao Xie. Sequential Predictive Conformal Inference for Time Series. ICML 2023.

**Questions:**

- How does DualRes differ theoretically or practically from conformal methods adapted to time series forecasting, and why were they omitted from the experiments?
- How robust is the method to model misspecification? For instance, if the normalized residuals are not i.i.d. as assumed, how does this affect the quality of the probabilistic forecasts?
- The multivariate extension seems to assume a diagonal conditional volatility structure. Does DualRes capture cross-variable dependencies at all, or would it require modeling the full covariance matrix $G$? Please clarify the intended scope of "spatial dependence".
- Could you provide more details on computational cost and scalability — for instance, the effect of the number of resamples $B$ on runtime and convergence? This would help assess practical feasibility compared to other probabilistic forecasting methods.

---

### Official Review · Reviewer_ikZk · 2025-10-29

**Soundness:** 3
**Presentation:** 3
**Contribution:** 3
**Rating:** 4
**Confidence:** 4

**Summary:**

The paper proposes DualRes that turns any mean-forecasting model into a probabilistic forecaster. It learns (i) a conditional mean $F$ and (ii) a conditional volatility $G$ via a log-squared–residual regression trick, then resamples normalized residual vectors at inference to synthesize predictive distributions—thereby avoiding Gaussian assumptions, handling conditional heteroskedasticity, and extending naturally to multivariate series (by resampling residual vectors). A consistency result (Theorem 1) shows the empirical CDF of normalized residuals converges to the true innovation distribution under i.i.d. innovations. Experiments on six univariate datasets and three multivariate settings report CRPS/MAEC/ES gains over DeepAR, DLinear, PatchTST, TimeMixer, VEC-LSTM, and TMDM.

**Strengths:**

- Simplicity & generality: drop-in on many forecasters; no need to design likelihoods.
- Beyond Gaussianity: residual-vector resampling naturally captures heavy tails and spatial dependence; histograms show non-Gaussian residuals.
- Empirical coverage & accuracy: frequent CRPS/MAEC/ES gains; narrower CIs in several cases.

**Weaknesses:**

- The optimization is a two-staged process instead of end-to-end.
- The method relies on i.i.d. innovations and accurate $F$ and $G$. Lack of analysis under innovation misspecification/auto-correlated noise.
- Not universally better. Some degradations (e.g., PatchTST on ETTh1 and M4-Hourly) suggest instability and warrant analysis/guardrails.
- Design choices underexplored. Only i.i.d. residual resampling is tested; alternatives (block/bootstrap variants), lag selection $q$, $s$, and diagonal-$G$ vs. full/low-rank forms are not ablated. (Sec. 6 notes compute as a limitation but lacks complexity/runtime tables.)

**Questions:**

- When does it fail? Provide diagnostics or conditions (on heteroskedasticity strength, residual skew/kurtosis, or base-model bias) predicting when DualRes helps vs. hurts; analyze the cases where CRPS worsens.
- Resampling scheme. Why i.i.d. resampling rather than block/wild/bootstrap variants for small-$T$ or residual dependence? Any empirical comparison? (This seems especially relevant for long-horizon rollouts.)
- Volatility model. Sensitivity to the diagonal-$G$ assumption and to errors in the log-squared residual regression; could a low-rank-plus-diag $G$ improve multivariate ES further?
- Compute & budget parity. Please report training/inference overhead (per backbone) and confirm equal training budgets for baselines vs. “+Ours.”

---

### Official Review · Reviewer_k9hX · 2025-10-31

**Soundness:** 1
**Presentation:** 1
**Contribution:** 2
**Rating:** 2
**Confidence:** 4

**Summary:**

This paper proposes a post-hoc approach to probabilistic forecasting where in addition to the conditional mean model, they fit a conditional model on the residuals and plug-it in a noise model that ultimately enables probabilistic forecasting. The algorithms is then tested against univariate and multivariate time series datasets with baselines that output point forecasts, and one baseline that is probabilistic by design.

**Strengths:**

- The fact that the approach is post-hoc and applicable on top of any point forecasts model is practical and flexible.

**Weaknesses:**

- The writing of the paper is convoluted (e.g. line 36), and even in the methods section (section 3) I find it hard to follow the authors logic.
- In the introduction, the authors mention diffusion-based and flow-matching baselines and criticize these as relying on "time series having Gaussian distribution" assumption, which to the best of my knowledge is not true for this kind of methods that are able to approximate any complex distribution. For instance Flow-matching allows this through the learning of the velocity field mapping samples from a standard Gaussian distribution to samples from the desired complex distribution (as can be seen in [1]).
- The literature that the authors are citing about probabilistic forecasting (mainly around diffusion and flow-matching models) is not complete. Other ways of doing probabilistic forecasting include: Quantile regression (as done in Chronos [2]), Parametric models (the most simple ones use a Gaussian distribution such as DeepAR, but more complex mixture based models exist such as Moirai [3]), and Bayesian methods (as in [4]).
- The comparison in Table 1 is not fair to me as 3 of the 4 considered baselines were originally trained for point forecasts and the comparison is solely done in terms of probabilistic metrics. The authors mention that for these baselines "their distributional indices are obtained through fitting a t-distribution to the predictive values", can you elaborate more on what does that mean exactly?

[1] Liu, Yong, et al. "Sundial: A family of highly capable time series foundation models." _arXiv preprint arXiv:2502.00816_ (2025).

[2] Ansari, Abdul Fatir, et al. "Chronos-2: From Univariate to Universal Forecasting." _arXiv preprint arXiv:2510.15821_ (2025).

[3] Woo, Gerald, et al. "Unified training of universal time series forecasting transformers." (2024): 53140.

[4] Benechehab, Abdelhakim, et al. "AdaPTS: Adapting Univariate Foundation Models to Probabilistic Multivariate Time Series Forecasting." _arXiv preprint arXiv:2502.10235_ (2025).

**Questions:**

- Although Theorem 1 is interesting in itself, I find that in its current formulation it doesn't add match value as in practice, we would never be in the case where the mean and residuals are fitted perfectly. Can the authors discuss or explore parallel formulations that can characterize the error in probability between the true distribution and the learned distribution, as a function of the estimation error of the mean and residuals? I believe such a result can help getting some insights about the expected error in practice, and would improve the paper.

---

### Official Review · Reviewer_JYvW · 2025-10-31

**Soundness:** 2
**Presentation:** 1
**Contribution:** 1
**Rating:** 2
**Confidence:** 4

**Summary:**

This paper introduces DualRes, a method designed to capture conditional heteroskedasticity and residual distributional information for probabilistic time series forecasting. The core of the approach involves training two separate components: one dedicated to modeling the conditional mean and the other for the volatility of the time series. The authors integrate DualRes with existing forecasting models and present experimental results to showcase the resulting performance improvements.

This paper's presentation is not clear to follow. Some mathematical formulations are confusing, as outlined in the weaknesses section.
Overall, this paper's quality is not ready for publishing. Based on the issues with clarity and presentation, particularly concerning the mathematical rigor, this paper's current quality is not yet ready for publication.

**Strengths:**

This paper focuses on probabilistic forecasting for time series, which can provide both point and distribution predictions.

In the experiment, the proposed method is applied to several existing models, which shows the potential compatibility of the proposed method.

**Weaknesses:**

The weaknesses are as follows.

1. Mathematical presentation lacks rigor. For instance, in line 100, x_{1:T} is defined as a d-dimensional vector, and thus the suffix 1:T is confusing.

In Eq.(1), why is the G(\cdots) defined as a diagonal matrix?

2. Algorithm 1 lacks sufficient details to justify the correctness and reasonability. For instance, it is unclear whether steps 1 and 2 are sequentially trained or collectively trained.

3. Sec.4 is specifically difficult to follow, and it is unclear what the authors aim to prove.

**Questions:**

Refer to the weakness section.

---

### Note · Authors · 2025-11-28

I have read and agree with the venue's withdrawal policy on behalf of myself and my co-authors.